# Whole blood stabilization for the microfluidic isolation and molecular characterization of circulating tumor cells

Keith H.K. Wong [1,2], Shannon N. Tessier[1,2], David T. Miyamoto[3,4], Kathleen L. Miller[1,2], Lauren D. Bookstaver[1,2], Thomas R. Carey[1,2], Cleo J. Stannard[1,2], Vishal Thapar[3,5], Eric C. Tai[3], Kevin D. Vo[3], Erin S. Emmons[3], Haley M. Pleskow[4], Rebecca D. Sandlin[1,2], Lecia V. Sequist[3,6], David T. Ting[3,6], Daniel A. Haber[3,6,7], Shyamala Maheswaran[2,3], Shannon L. Stott[1,3,6] & Mehmet Toner[1,2]

Precise rare-cell technologies require the blood to be processed immediately or be stabilized with fixatives. Such restrictions limit the translation of circulating tumor cell (CTC)-based liquid biopsy assays that provide accurate molecular data in guiding clinical decisions. Here we describe a method to preserve whole blood in its minimally altered state by combining hypothermic preservation with targeted strategies that counter cooling-induced platelet activation. Using this method, whole blood preserved for up to 72 h can be readily processed for microfluidic sorting without compromising CTC yield and viability. The tumor cells retain high-quality intact RNA suitable for single-cell RT-qPCR as well as RNA-Seq, enabling the reliable detection of cancer-specific transcripts including the androgen-receptor splice variant 7 in a cohort of prostate cancer patients with an overall concordance of 92% between fresh and preserved blood. This work will serve as a springboard for the dissemination of diverse blood-based diagnostics.

[1] BioMEMS Resource Center, Center for Engineering in Medicine, Massachusetts General Hospital, Harvard Medical School, Boston, MA 02114, USA. [2] Department of Surgery, Massachusetts General Hospital Harvard Medical School Boston, MA 02114, USA. [3] Cancer Center, Massachusetts General Hospital Harvard Medical School Boston, MA 02114, USA. [4] Department of Radiation Oncology, Massachusetts General Hospital Harvard Medical School Boston, MA 02114, USA. [5] Department of Pathology, Massachusetts General Hospital Harvard Medical School Boston, MA 02114, USA. [6] Department of Medicine, Massachusetts General Hospital Harvard Medical School Boston, MA 02114, USA. [7] Howard Hughes Medical Institute, Chevy Chase, MD 20815, USA. Keith H.K. Wong and Shannon N. Tessier contributed equally to this work. Correspondence and requests for materials should be addressed to S.L.S. (email: sstott@mgh.harvard.edu) or to M.T. (email: mtoner@hms.harvard.edu)

Peripheral blood contains a tremendous amount of cellular and molecular information relating to the entire body, and the investigation of blood-borne cells is of broad significance to clinical medicine and basic research. In particular, recent innovations in rare-cell and molecular technologies are rapidly advancing our ability to isolate and characterize circulating tumor cells (CTCs) for the noninvasive detection and monitoring of cancer. CTC-based liquid biopsy technologies have now expanded into a wide spectrum of applications in precision oncology, including predictive biomarker discovery, understanding mechanisms of drug resistance and metastasis, and personalized testing of drug efficacy[1–3].

However, similar to any procedures involving live tissues, blood degradation during the handling of samples and laboratory manipulations imposes practical constraints and represents a major roadblock to the translation of modern liquid biopsy technologies. Once removed from its native environment, a host of degenerative processes including hemolysis, platelet activation, cytokine and oxidative bursts, and neutrophil extracellular trap formation[4] inflict collateral damage to the entire blood specimen.

These problems are exacerbated by the extreme rarity and fragility of CTCs[5,6] not only because the target cells are buried in such a hostile environment but also due to the breakdown of stringent rare-cell sorting mechanisms when challenged with disintegrated blood cells, extracellular DNA, as well as altered cellular morphology and marker expression[7]. Controlled studies using spiked tumor cells have documented a >60% loss in CTC yield within 5 h of blood draw[8], and significant RNA degradation occurs within 2–4 hours[9–11]. In clinical studies where short-term storage for 3–4 h is common, ~ 40% of isolated single cells failed RNA quality control for profiling[12,13]; within 12 h, RNA degradation could be found in 79% of cells[14]. Although modern transfusion medicine has established protocols for the banking of purified blood components, these techniques fall short of preserving whole blood for rare-cell applications. For instance, cryogenic storage requires high concentrations of toxic cryoprotectants (e.g., 40% glycerol or dimethylsulfoxide) with complicated slow-freezing and washing protocols that are not practical for routine clinical workflow and quality control. Conditions optimized for one cell type are not necessarily beneficial to others

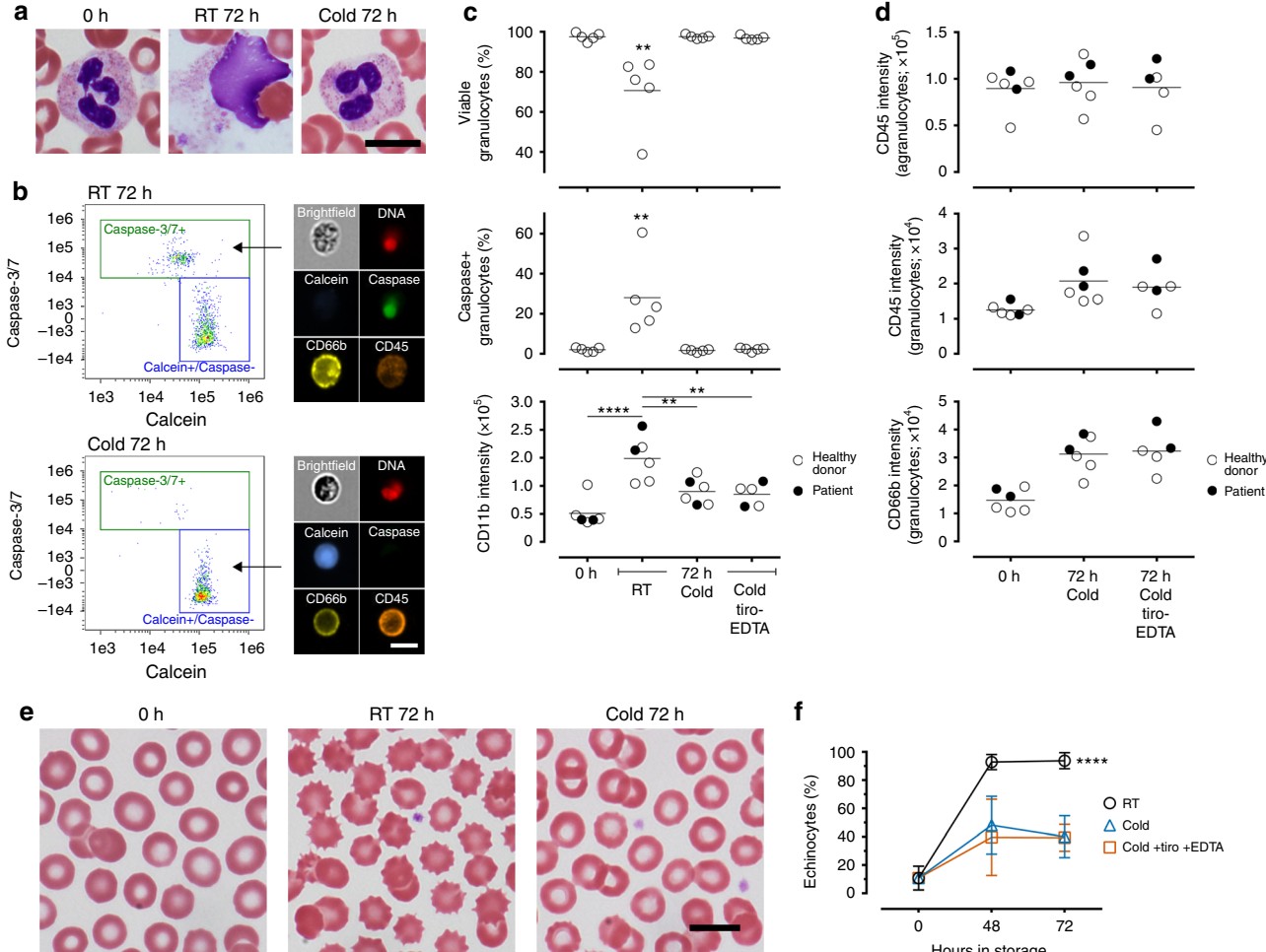

**Fig. 1** Hypothermic storage of ACD-anticoagulated blood preserves the viability and integrity of diverse cell types. **a** Representative images of neutrophils in fresh (0 h) and stored (72 h) blood. Storage in RT leads to cell death and release of nuclear materials, whereas storage in 4 °C (cold) preserves cellular integrity and the distinct segmented nuclear morphology. **b** Imaging flow cytometry for the quantification of viable (calcein+/caspase−) and apoptotic (caspase+) leukocytes as well as surface marker expression. **c** Percentages of viable and apoptotic granulocytes (CD45+/CD66+), and their expression of CD11b in fresh and stored blood (**p < 0.01; ****p < 0.0001; one-way ANOVA followed by Tukey's post test). **d** Surface expression of CD45 and CD66b on agranulocytes (CD45+/CD66b−) and granulocytes. **e** Representative images of RBCs in fresh and stored blood. Echinocytes are identified by the distinct spiculations as shown in RT-stored blood. **f** Percentages of echinocytes as a function of storage (****p < 0.0001; two-way ANOVA followed by Tukey's post test). **c**, **d**, **f** The platelet stabilization cocktail treatment (tiro-EDTA) does not affect cell viability, activation, marker expression, or echinocyte formation. All scale bars represent 10 μm. Error bars represent SD

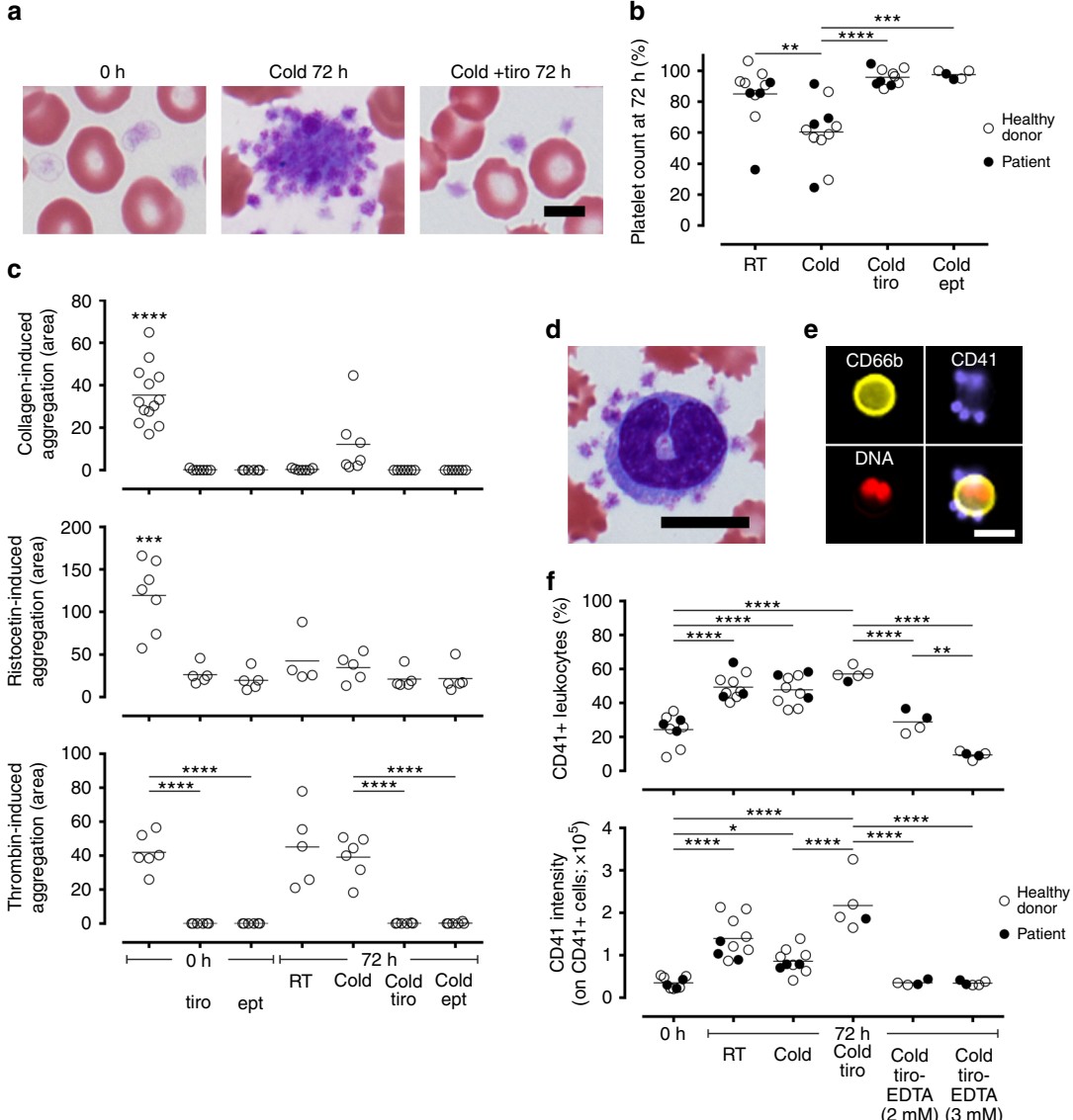

**Fig. 2** Platelet stabilization with GPIIb/IIIa inhibitors and calcium chelation. **a** Representative images of platelets demonstrating cold-induced aggregation, which is inhibited by tirofiban (tiro; 0.5 µg mL$^{-1}$). **b** Changes in platelet count as a result of storage in the presence of tirofiban or eptifibatide (ept; 50 µg mL$^{-1}$), or without any inhibitors. A decrease in count relative to 0 h indicates aggregation. **c** Platelet aggregation induced by collagen, ristocetin, and thrombin in both fresh and stored blood (with or without inhibitors) measured by impedence aggregometry. **d** A representative image of platelet cloaking in stored blood. **e**, **f** Platelet cloaking quantified by imaging flow cytometry. **f** CD41+ leukocytes are cells that are positive for at least one platelet. CD41 intensity quantifies the extent of platelet attachment on these cells. Scale bar in **a** represents 5 µm. Other scale bars represent 10 µm. *$p < 0.05$; **$p < 0.01$; ***$p < 0.001$; ****$p < 0.0001$ (one-way ANOVA followed by Tukey's post test)

—for example, low temperatures used for red blood cell (RBC) preservation (2–6 °C) lead to spontaneous platelet activation, which causes nonspecific binding and aggregation[15,16]. Alternatively, commercial platforms, including CellSearch, the only Food and Drug Administration (FDA)-cleared CTC platform, rely on fixatives to stabilize whole blood for up to 96 h to accommodate specimen storage and transportation for multi-center studies[17–19]. The tradeoff, however, is that fixation not only sacrifices cell viability but also degrades RNA[20] due to chemical crosslinking, fragmentation, and chemical modifications[21]. Preservation of whole blood in an unaltered state is therefore critical for acquiring clinically actionable information such as gene expression profiling as well as establishing ex vivo cultures and xenograft models[3].

Here we present a method that preserves whole blood in an unfixed, viable state for up to 72 h for rare-cell sorting and RNA

profiling. We focus on RNA because CTC transcriptomics is invaluable to evaluate tumor heterogeneity and to define signaling pathways relevant to cancer progression and drug resistance[13]. Moreover, mRNA splice variants have been identified as potential predictive biomarkers of treatment response[22]. We test our preservation protocol using the recently developed microfluidic technology, the CTC-iChip, which isolates CTCs in an unbiased manner through high-throughput depletion of hematologic cells[23,24]. We first identify hypothermic storage conditions that preserve the integrity and surface epitopes of diverse hematologic cell types. To counter cooling-induced platelet activation, we employ glycoprotein IIb/IIIa (GPIIb/IIIa) inhibitors to enable clot-free microfluidic processing and apply a brief calcium chelation treatment to reverse nonspecific platelet cloaking. This approach enables the efficient sorting of rare CTCs from blood that has been preserved for up to 72 h while retaining cell viability

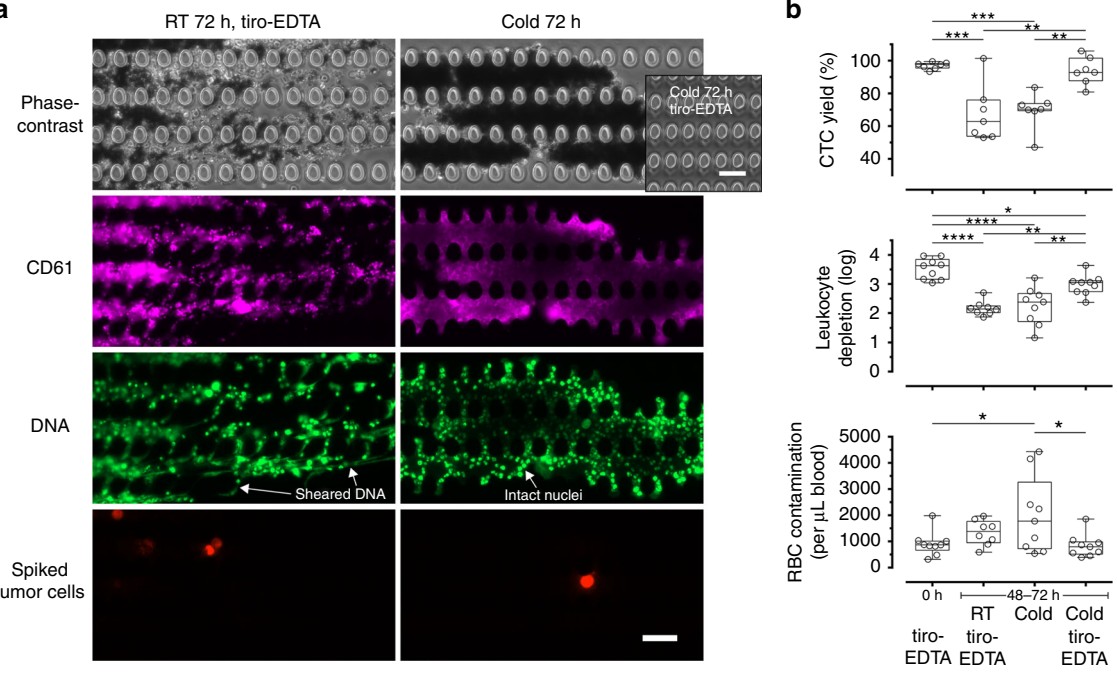

**Fig. 3** Functional operation of microfluidic CTC isolation requires stabilized whole blood. **a** Representative images of the micropost array that performs size-based sorting (debulking) in the CTC-iChip. Blood storage in room temperature, even if treated with tiro-EDTA, results in aggregates that contain sheared DNA consistent with cell death and extracellular trap formation. Cold storage without tiro-EDTA leads to clots that contain densely packed platelets (CD61 staining) and intact cells. In both cases, rare cells are trapped within the aggregates. Cold storage with tiro-EDTA consistently permits clean processing (inset). **b** CTC isolation performance of the CTC-iChip in different storage conditions. All scale bars represent 50 μm. Box-and-whiskers plots show median, interquartile range, maxima, and minima. *$p < 0.05$; **$p < 0.01$; ***$p < 0.001$; ****$p < 0.0001$ (one-way ANOVA followed by Tukey's post test)

and intact, high-quality RNA for molecular profiling. Together, this work overcomes a critical logistical hurdle in the isolation of live cells from whole blood for clinical and scientific investigations.

## Results

**Defining storage conditions that preserve whole blood cells**. In modern blood banking and immunologic testing, whole blood is held at ambient temperature before processing into sub-components—ideally within several hours—for storage or functional assays[25,26]. Because granulocytes represent the most abundant and short-lived leukocytes[27], we analyzed their viability using imaging flow cytometry to benchmark storage conditions (Fig. 1a, b). We found that hypothermic storage (cold; 4 °C) using the anticoagulant Acid Citrate Dextrose (ACD; Supplementary Fig. 1) had a clear benefit on granulocyte preservation: their viability (calcein+/caspase−) after 72 h of cold storage (97.5 ± 0.9%, mean ± SD throughout the text unless specified, $n = 5$; Fig. 1c) was no different from fresh control samples (0 h; 97.5 ± 2.0%, $n = 5$; Fig. 1c), while room-temperature storage (RT; ~ 22 °C) resulted in ~ 30% cell death (Fig. 1c). The majority of non-viable cells (96%) stained positive for caspase-3/7 activity, suggesting apoptosis as the primary mechanism of cell death. Cold storage was also associated with a lower degree of granulocyte activation measured by the expression of CD11b (Fig. 1c). Further, the pan-leukocyte marker CD45 and granulocyte marker CD66b, both widely used in cell isolation assays[23,24,28], were well-retained (Fig. 1d). Similarly, RBC aging quantified by echinocyte formation was greatly inhibited by cold storage compared to RT (Fig. 1e, f). These results demonstrate that cold storage of ACD-anticoagulated blood sufficiently maintains cellular morphology,

integrity, and surface epitope stability of diverse hematologic cell types.

**Platelet activation during hypothermic blood storage**. A critical issue in exposing platelets to hypothermic temperatures is their spontaneous activation (Fig. 2a), which not only has biological impacts on other cell types but also leads to undesired clotting that can result in the failure of microfluidic blood processing (Fig. 3a). To understand the effects of storage temperature on platelets, we quantified the loss of single platelets and performed whole blood impedance aggregometry to characterize their coagulation response. Blood storage for 72 h at RT led to a ~ 15% decrease in platelet count (Fig. 2b), and this decrease was significantly higher with cold storage (~ 40% drop; Fig. 2b). We then tested the functional response of platelets when challenged with collagen type I, ristocetin, and thrombin. These agonists initiate platelet activation via different pathways, with thrombin being the key serine protease that catalyzes fibrin polymerization in the final common pathway. Both RT and cold storage resulted in decreased response to collagen and ristocetin (Fig. 2c). Response to thrombin, however, was retained under both storage conditions (Fig. 2c). These observations on platelet functions in whole blood are consistent with early studies on the preservation of platelet concentrates[29–31].

**Platelet stabilization for hypothermic blood preservation**. The observation that platelets remain fully responsive to thrombin suggests that thrombus formation mechanisms remain intact. In cardiovascular medicine, specific GPIIb/IIIa inhibitors are indicated to prevent blood clotting in ischemic events[32]. We tested two such inhibitors, tirofiban (0.5 μg mL$^{-1}$) and eptifibatide (50

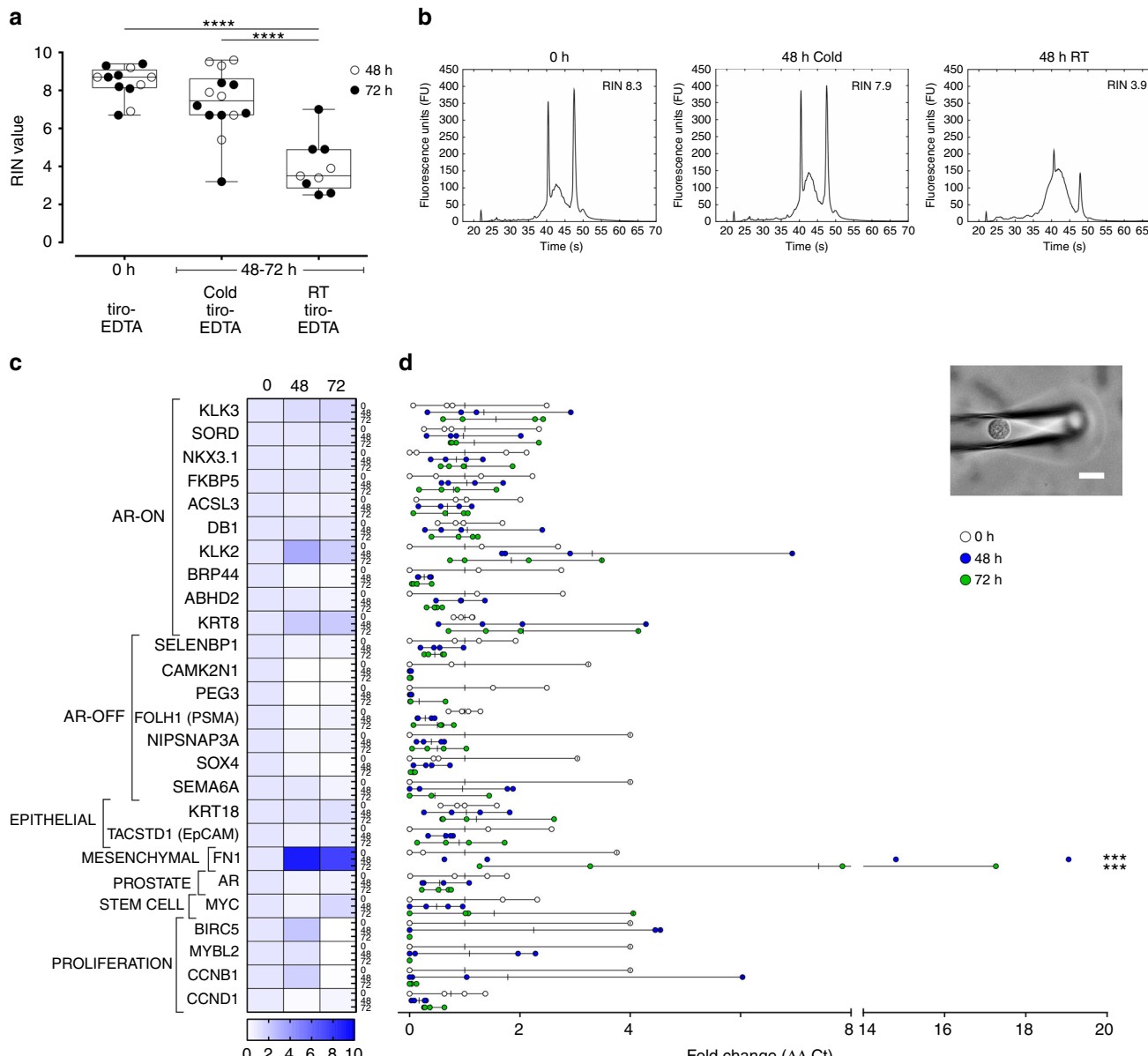

**Fig. 4** LNCaP cells spiked into healthy donor blood and isolated from the CTC-iChip contain intact RNA for molecular profiling after up to 72 h of preservation. **a** RIN values of isolated LNCaP cells from fresh and stored blood (****$p < 0.0001$; one-way ANOVA followed by Tukey's post test). **b** Representative electropherograms showing the size distribution traces of total RNA. Cells isolated from RT-stored blood contain highly fragmented RNA. In contrast, cells isolated from fresh and cold-stored blood contain intact RNA with distinct 28S/18S ribosomal peaks. **c** Heatmap and **d** fold change (relative to 0 h) of a panel of 26 prostate cancer genes in single LNCaP cells (picked from the CTC-iChip product; inset) comparing 0-, 48-, and 72-h cold storage (***$p < 0.001$; two-way ANOVA followed by Dunnett's post test). Heatmap in **c** represents the average expression across single cells, and data points in **d** represent individual cells. Scale bar represents 20 μm. Box-and-whiskers plots show median, interquartile range, maxima, and minima

μg mL$^{-1}$), and found that both of them completely inhibit the decrease in platelet count after cold storage of whole blood (Fig. 2b). Importantly, they completely inhibited platelet aggregation induced by thrombin in both fresh and cold-stored blood (Fig. 2c). Next, we examined platelet cloaking (Fig. 2d), which masks other cells and interferes with their immunocapture integral to cell isolation mechanisms. Using imaging flow cytometry to quantify platelet–leukocyte adhesion (Fig. 2e), we found that storage leads to a 2-fold increase in the number of platelet-positive (CD41+) leukocytes, and the extent of platelet attachment (CD41 intensity) on these leukocytes concomitantly increased (4- and 2.5-fold increase for storage in RT and cold compared to 0 h, respectively; Fig. 2f). The addition of tirofiban, although not affecting the number of CD41+ leukocytes, led to

increased CD41 intensity on these leukocytes (Fig. 2f) presumably because tirofiban frees up single platelets to interact with other cells. Because platelet–leukocyte interaction is mediated by a variety of calcium-dependent selectins and integrins[33], we reasoned that chelating divalent ions would reverse such binding. We found that a brief treatment with EDTA post-storage (2–3 mM for 15 min) was sufficient to release these platelets (Fig. 2f). This short EDTA treatment (up to 75 min) did not alter the expression of CD45, CD66b, or CD11b (Fig. 1c, d), nor did it impact granulocyte viability or echinocyte formation (Fig. 1c, f). This treatment provides a reasonable time frame for the micro-fluidic processing of blood samples.

Together, our results indicate that whole blood storage leads to platelet aggregation and platelet–leukocyte adhesion.

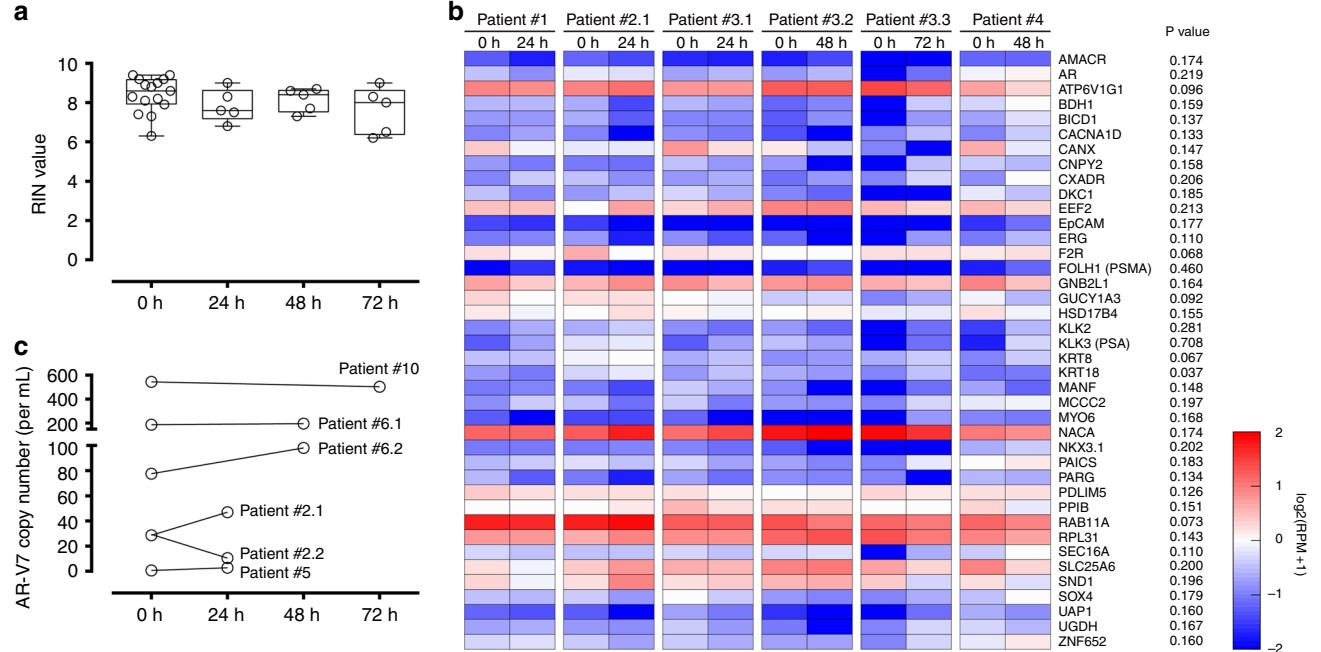

**Fig. 5** CTCs isolated from prostate cancer patient blood contain intact RNA for RNA sequencing and Droplet Digital PCR after whole blood preservation. **a** RIN values of CTCs isolated from fresh and cold-stored blood (total 15 blood draws from 8 independent patients). **b** Scaled heatmap of log2-transformed normalized reads (RPMs) for four prostate cancer patients. For patient #3, data are available from three different draw dates stored for different durations (24, 48, and 72 h; denoted #3.1, #3.2, and #3.3, respectively). Statistical analysis comparing fresh vs. cold-stored blood (6 blood draws) found no significant differences in the mean RPMs for any of the 40 genes depicted in **b**, except for *KRT18* ($p = 0.037$, paired *t*-test). **c** The mRNA copy number of *AR-V7* detected by Droplet Digital PCR (expressed as per mL of blood processed). Box-and-whiskers plots show median, interquartile range, maxima, and minima

Glycoprotein IIb/IIIa inhibitors completely abrogate aggregation due to storage and thrombin stimulation, and a short-term treatment with EDTA is sufficient to release platelets from platelet-bound leukocytes. We have confirmed that GPIIb/IIIa inhibition is necessary and sufficient to prevent clogging in the microfluidic processing of cold-stored blood (Supplementary Fig. 2), demonstrating the specificity of this interaction.

**Efficient microfluidic sorting of CTCs from preserved blood**. Isolation of CTCs for RNA analyses is currently performed within 4 h of blood collection to minimize degradation[12,13,34,35]. We therefore hypothesized that the combined benefits of hypothermic blood preservation and platelet stabilization would enable the microfluidic isolation of rare CTCs after extended storage (48–72 h). We tested CTC sorting performance using our latest microfluidic technology, the CTC-iChip[24], to study the effects of platelet stabilization using tirofiban with post-storage EDTA treatment (denoted tiro-EDTA) as well as storage temperature (RT vs. cold). For fresh blood, we found that tiro-EDTA greatly increased the purity of the CTC product (Supplementary Fig. 3). For stored blood (48–72 h), conditions that fail to either preserve blood cells (i.e., RT w/ tiro-EDTA) or stabilize platelets (i.e., cold w/o tiro-EDTA) resulted in severely compromised microfluidic processing due to distinct mechanisms. In RT storage, aggregates that clog the device contain abundant sheared DNA strands (Fig. 3a), consistent with cell death and neutrophil extracellular trap formation. Cold storage without tiro-EDTA led to densely packed aggregates consisting of platelets and intact cells (Fig. 3a). In both conditions, spiked tumor cells were trapped inside these aggregates. Clean, clog-free microfluidic processing of stored blood was only possible using cold-preserved blood that was treated with tiro-EDTA (inset in Fig. 3a). In terms of key metrics of CTC sorting performance (Fig. 3b), cold storage

with tiro-EDTA achieved a CTC yield of $93.8 \pm 8.4\%$ ($n = 7$), which is significantly higher than other storage conditions and not different from fresh blood. In addition, leukocyte depletion remained high at $2.97 \pm 0.35$-log ($n = 9$), compared to $2.17 \pm 0.25$-log for RT storage (w/ tiro-EDTA; $n = 8$) and $2.24 \pm 0.63$-log for cold storage (w/o tiro-EDTA; $n = 9$). These depletion levels equate to ~6800 contaminating leukocytes per mL of blood for cold storage (w/ tiro-EDTA), compared to ~39,000 and ~81,000 for RT (w/ tiro-EDTA) and cold (w/o tiro-EDTA), respectively. Similarly, RBC contamination was minimal and not different from fresh blood (Fig. 3b).

**Molecular profiling of spiked tumor cells from stored blood**. We next assessed whether CTCs isolated from preserved blood contain intact RNA compatible with tumor molecular profiling. We spiked LNCaP cells (a human prostate cancer cell line) into healthy donor blood, applied the platelet stabilization treatment, and processed fresh or stored blood using the CTC-iChip. We then evaluated bulk RNA quality (RNA integrity number, RIN) in the cells recovered from the CTC-iChip and analyzed mRNA expression in individual cells using quantitative real-time PCR (RT-qPCR). The mean RIN value of spiked tumor cells isolated from cold-preserved (48–72 h) blood was $7.4 \pm 1.7$ ($n = 14$), which is not different from those isolated from fresh blood ($8.4 \pm 0.9$, $n = 12$; $p = 0.16$, one-way analysis of variance (ANOVA) followed by Tukey's post test; Fig. 4a). Representative electropherograms displayed distinct 28S/18S ribosomal peaks indicating minimal RNA degradation (Fig. 4b). In contrast, cells isolated from RT-stored blood contained significantly degraded RNA (RIN = $4.0 \pm 1.4$, $n = 9$; $p < 0.0001$ compared to 0 h and cold storage, one-way ANOVA followed by Tukey's post test; Fig. 4a), characterized by decreased 28S rRNA peak heights (28S rRNA is

typically degraded more quickly than 18S rRNA) and increased prevalence of RNA fragments towards the fast region (Fig. 4b).

Since mRNA is particularly sensitive to degradation, we further characterized the effect of cold preservation on gene expression signatures. We chose to perform single-cell RT-qPCR (Fig. 4c, d) because bulk mRNA quantification that includes contaminating leukocytes masks the heterogeneity of tumor cells, which may be differentially susceptible to storage. Single LNCaP cells isolated from fresh and cold-preserved blood were micromanipulated (inset in Fig. 4d) and analyzed for the relative expression of a panel of 26 genes relevant to prostate cancer (Fig. 4c, d) as previously described[23]. Leukocyte-specific markers were also included to ensure the detection of bona fide cancer signatures. No statistically significant differences were detected in levels of the 26 genes expressed in LNCaP cells isolated from fresh blood and 48- or 72-h-preserved blood, with the exception of FN1 (fibronectin; $p = 0.0001$, two-way ANOVA followed by Dunnett's post test).

**Molecular profiling of patient CTCs from preserved blood**. To evaluate the compatibility of our protocol with clinical samples, we verified bulk RNA quality and performed next-generation RNA sequencing on CTCs isolated from a cohort of prostate cancer patients (total 12 independent patients). We applied the tiro-EDTA protocol and stored patient blood in cold for 24–72 h prior to CTC-iChip processing. For bulk RNA quality, we found no differences in RIN values for CTCs isolated from fresh and preserved patient blood (Fig. 5a), nor did we observe any trend with respect to storage duration. We also performed control experiments in which patient blood was stored and processed without tiro-EDTA (Supplementary Fig. 4). As expected, platelet stabilization was essential for maintaining RNA quality in patient-derived CTCs, providing further validation of its utility in clinical specimens.

We then tested whole-transcriptome profiles using RNA-Seq of CTCs isolated from patients with advanced metastatic castration-resistant prostate cancer. Genes known to be upregulated in the prostate were selected from published literature[36] and the specificity for prostate tissue vs. whole blood was verified using the GTEx Portal. Figure 5b shows a scaled heatmap of log2-transformed normalized reads (i.e., reads per million, RPM) for four prostate cancer patients; for patient #3, data were available from three different blood draws with different storage durations (24, 48 and 72 h). No significant differences were evident between fresh and preserved blood (six blood draws; paired t-test) for any of the 40 genes except for KRT18 ($p = 0.037$; Fig. 5b). Select genes in certain patients showed a trend toward increased expression. Further studies are required to understand the mechanisms driving this upregulation and the associated clinical implications.

We extracted the top 11 genes with the most stringent prostate cancer specificity and evaluated their binary concordance between fresh and preserved blood (Supplementary Table 1). For patients #1, #2.1, #3.1, and #4, 100% of these genes detected in CTCs from fresh blood were also detected after storage. For patient #3.2, 9 out of 11 genes were concordant. Finally for patient #3.3, 7 of the 11 genes were concordant between fresh and preserved blood. In this particular blood draw, however, the overall low numbers of cancer-specific reads may have contributed to the higher variability considering the heterogeneity of rare CTCs.

**Detection of androgen-receptor splice variant 7 transcripts**. A particularly important and clinically relevant biomarker in prostate cancer is the androgen-receptor splice variant 7 (AR-V7), which encodes an androgen receptor (AR) protein isoform that constitutively activates the AR pathway in the absence of androgens. The presence of AR-V7 mRNA in CTCs predicts poor response to inhibitors of AR signaling in metastatic castration-resistant prostate cancer patients[22]. Because AR-V7 mRNA is present in much lower quantities than full-length AR mRNA (10- to 100-fold)[13,22], we used Droplet Digital PCR (ddPCR) to more accurately quantify its level of expression. In a cohort of 7 independent patients (total 9 blood draws), we detected AR-V7 transcripts in 4 patients across 6 blood draws with 100% concordance between fresh and preserved blood (Fig. 5c and Supplementary Table 1). Three other patients with undetectable AR-V7 at 0 h did not show any positivity after blood storage. Together with results from RNA-Seq, we obtained an overall binary concordance of 92.0% in terms of cancer-specific signal detection between fresh and preserved blood across 10 patients and 14 independent blood draws (Supplementary Table 1).

## Discussion

Recent innovations in rare-cell technologies provide unprecedented opportunities for the noninvasive management of cancer. Although the FDA-cleared CellSearch platform enables CTC enumeration and provides prognostic information, development of more advanced applications that rely on detailed molecular analysis of CTCs has been limited by the requirement to process fresh blood within several hours of blood draw. Of note, many of these applications, including transcriptional analyses and cell culture, are not possible with circulating tumor DNA or other cell-free liquid biopsy approaches. In this study, we employed a three-pronged approach—hypothermic preservation, GPIIb/IIIa inhibition, and post-storage calcium chelation—to extend the lifespan of whole blood for up to 72 h for microfluidic blood processing. This non-fixative method effectively preserves cell viability in whole blood and enables the isolation of intact CTCs with high-quality RNA for transcriptomic analyses. Importantly, this method makes feasible the real-world clinical implementation of recently discovered CTC-based RNA biomarkers with the potential to inform clinical decisions, such as AR-V7 in prostate cancer and liver-specific transcripts for the early detection of hepatocellular carcinoma[37].

Blood stabilization for rare-cell applications—the identification and isolation of one in a billion blood cells—requires an approach fundamentally different from existing modalities in blood banking. The CTC-iChip used in this study represents one of the most rigorous tests of blood quality due to its reliance on precise continuous-flow sorting of a highly complex fluid: it first "debulks" whole blood using hundreds of thousands of microposts that remove RBCs and platelets; followed by inertial focusing that aligns nucleated cells into a single file; and finally depletes pre-tagged leukocytes by magnetophoresis, leaving untouched CTCs in the final product. As such, thrombi or extracellular DNA (Fig. 1a) would physically clog channels and trap target cells (Fig. 3a), and in severe cases lead to systematic device failure. Morphological deteriorations such as echinocytosis (Fig. 1e) and cellular aggregation lead to ineffective size-based separation. These together with surface marker degradation and platelet cloaking (Fig. 2d, e)—both of which hinder immunoaffinity-based targeting—lead to significant contamination of hematologic cells in the CTC product (Fig. 3b). These challenges are by no means unique to select CTC technologies; the very same sorting mechanisms are frequently employed in a broad range of microfluidic devices for CTCs[38–40] and beyond, including neutrophil expression profiling[28], CD4+ cell detection for HIV monitoring[41], and rare fetal cell isolation from maternal blood[42], to name just a few examples.

In light of these challenges, we sought to develop a "universal" approach to blood stabilization—one that maintains whole blood in its viable, minimally altered state. To this end, we focus our efforts on hypothermic storage since it is a widely employed strategy in organ preservation and will be compatible with established cold chain protocols for transportation. Its primary working principle is to reversibly slow down cellular processes—for every 10 °C decrease in temperature, biological reactions are roughly halved. For example, RBCs suffer from "storage lesions" during which glucose, 2,3-diphosphoglycerate and ATP are depleted[43]. Accordingly, cooling to 2–6 °C greatly extends the availability of metabolic substrates and thus length of storage to 42 days for transfusion. Our results show that hypothermia exerts clear benefits on both RBC and leukocyte preservation for up to 72 h, accompanied by the maintenance of surface marker expression and inhibition of leukocyte activation (Fig. 1). We also note that alterations to RBC rheology—an important consideration for microfluidic processing—is minimal within this storage time frame[44,45]. The comprehensive nature of this approach is further demonstrated by our ability to obtain accurate complete blood counts (CBCs) after up to 72 h of cold preservation (Supplementary Fig. 5). CBC is one of the most important diagnostic tests that detect a wide range of disorders, and these counts directly impact the use of 136 items on the World Health Organization's (WHO) Model List of Essential Medicines[46]. Notably, our achievement in stabilizing surface marker expression and blood counts for up to 72 h surpasses the recommended storage duration set by the WHO by three- and sixfold, respectively[47]. The impact of these results on healthcare is significant; an estimated 46–68.2% of errors in laboratory testing are associated with the pre-analytical phase[48], in which inadequate sample collection, handling, transportation, and storage are thought to be the major contributors to error[49]. Indeed, it has been reported that up to 54% of missing test results can be attributed to hemolysis alone[50].

While the protective effects of hypothermia in biostabilization are well established, cooling-induced activation of platelets[15,16] limits their storage temperature to a narrow range of 20–24 °C. In initial experiments, we tested a range of antiplatelet agents, including acetylsalicylic acid (0.5–2 mM) and clopidogrel (10–100 μg mL$^{-1}$), as well as a p-selectin inhibitor KF 38789 (5–40 μg mL$^{-1}$), but found no functional benefits in our microfluidic assays. Our findings that stored whole blood remains fully responsive to thrombin stimulation (Fig. 2c) led us to target GPIIb/IIIa, which is the most abundant receptor on platelets and mediates fibrinogen binding during thrombus growth[51]. As our data show, the GPIIb/IIIa inhibitors tirofiban and eptifibatide completely abolish thrombin-induced aggregation in both fresh and stored blood (Fig. 2), and are extremely effective in microfluidic blood processing (Supplementary Fig. 2). Another unwanted effect of blood storage is platelet cloaking, a phenomenon resembling platelet satellitism in autoimmune conditions[52]. We found that ion chelation effectively unmasks these leukocytes (Fig. 2) and thereby enable their efficient tagging and microfluidic separation (Fig. 3b). Altogether, by stabilizing platelets in cold-preserved whole blood, CTCs can be readily isolated as if the blood specimen is freshly obtained with no modifications needed in terms of microfluidic operation. These benefits are reflected by the excellent yield (94%) and purity of the CTC product (Fig. 3b), requirements that are critical for molecular analyses.

Transcriptomic profiling holds immense promise in precision oncology given the rapid advancement and increasing affordability of next-generation sequencing technologies. RNA, however, is an extremely labile biomolecule. According to the WHO, the recommended storage duration of blood for RNA analysis by PCR amplification is <2 h[47]. This is consistent with CTC studies, which reported RNA degradation within 2–4 h[9–12]. We hypothesized that hypothermic preservation would decrease the activity of degradative pathways in CTCs and preserve their molecular signatures similar to what has been observed in other tissues[53,54]. Our data show that the RIN values from both spiked tumor cells and CTCs from prostate cancer patients are preserved after up to 72 h in cold storage (Figs. 4a and 5a). Since RIN values are largely influenced by rRNA, we also aimed to ensure that mRNA remains stable for cancer detection. Indeed, expression profiling of single tumor cells using a panel of 26 genes showed that there were no significant differences after 48 and 72 h of cold storage (Fig. 4c, d), with the exception of fibronectin. We postulate that this increase in extracellular matrix gene expression reflects a compensatory response in a non-adherent environment. Finally, we found that RNA integrity and mRNA are also preserved in clinical specimens. This confirmation is especially important given that cancer patients receive complex radiation and chemotherapy regimens that alter the hematopoietic system, and may result in more activated blood cells that negatively impact blood stabilization. Despite these additional challenges, our data demonstrate that RNA obtained from patient CTCs are of excellent quality after up to 72 h of storage (Fig. 5a), and that accurate molecular information can be generated by RNA sequencing with only 1 out of 40 genes being significantly different. We note, however, that a much larger cohort of patients will be required to perform equivalence testing—that is, to test whether the differences are within margins that can be considered "identical" in terms of clinical outcome[55]. Nevertheless, many diagnostic markers are clinically relevant based on a binary scale of "present or absent". Such is the case for AR-V7, which predicts resistance to AR inhibitors[22] and these patients may respond better to taxane chemotherapy[56]. In our patient cohort, we found a 100% binary concordance in AR-V7 detection and an overall concordance of 92% for all cancer-specific transcripts (Fig. 5b, c and Supplementary Table 1). In summary, the ability to stabilize whole blood for 72 h for RNA-based diagnostics effectively extends the time limit set by WHO standards by 36-fold. The reliable detection of AR-V7 mRNA from preserved blood may be immediately applicable to large-scale clinical trials for informing treatment decisions.

Moving forward, an extremely exciting area in precision oncology is the establishment of patient-specific CTC cultures and xenograft models for drug susceptibility testing[57–59]. The lack of robust methods to preserve viable CTCs is a major roadblock towards this Holy Grail in liquid biopsy. In our preliminary experiments, we found that spiked tumor cells in blood remain highly viable (>80%) after 72 h of hypothermic preservation (Supplementary Fig. 6); these encouraging data are also in line with our results on RNA quality and gene expression. Although more extensive studies are needed to characterize patient CTCs of diverse tumor types, our method opens up immediate opportunities in optimizing culture strategies for these novel rare cells.

By virtue of its universal nature in preserving whole blood, our approach is generally applicable to any cell isolation protocols for any downstream assays. This approach thus stands in stark contrast to existing methodologies, which focus on target applications at the expense of others. Formaldehyde-based fixation—although adequate in preserving a subset of surface epitopes—changes the rheology of blood and clogs isolation devices[60] in addition to degrading RNA. PAXGene (PreAnalytiX) is a blood collection tube that contains fixatives and additional RNA-stabilizing agents, and has been used for PCR-based detection of cancer-specific transcripts[61]. However, this tube is incompatible with any cell enrichment methods due to cell lysis and significantly decreases the yield of RNA[62]. The AdnaTest (QIAGEN)

uses a non-fixative preservative that stabilizes whole blood for up to 24 h and enables RNA detection from positively enriched CTCs[63]. In addition to the short stabilization window, this assay is only optimized for end-point PCR. Recently, a study explored the use of a proprietary sugar-based, non-fixative solution to stabilize whole blood for up to 7 days[64]. However, the authors did not quantify any metric of isolation efficiency (e.g., CTC yield and purity), nor did they perform any molecular characterization. Most importantly, despite the tremendous progress of microfluidic technologies that are capable of isolating blood cells at ever-increasing efficiencies[23,28,38–42,65], no RNA-stabilizing methods have demonstrated compatibility with these devices. Taken together, our achievements in enabling the microfluidic isolation of unfixed CTCs days after blood collection for transcriptomic profiling represent the most challenging benchmarks to date.

In conclusion, we have developed a "universal" protocol for whole blood stabilization and demonstrated its effectiveness in one of the most demanding and high-impact applications in precision oncology. Additional studies are warranted to more completely characterize whole-transcriptome alterations as a result of preservation, and to what extent they can be stabilized through other means such as further cooling (e.g., non-freezing sub-zero temperatures) or metabolic depression[66]. We note that our protocol offers a multitude of advantages in terms of clinical translation and regulatory approval. First, the required materials including GPIIb/IIIa inhibitors are already approved for clinical use and can be readily obtained at low costs. Second, the implementation of this protocol is straightforward, and no modifications of isolation assays are necessary. Finally, cold chain protocols are well established for the shipping of pharmaceuticals and biologics; compared to ambient storage, low temperatures are easily regulated and significantly lower the risk of bacterial contamination—an issue that necessitates costly screening on platelets. With further validations and benchmarking, we expect broad applicability of this protocol to a wide array of blood-based diagnostics and live-cell assays, for instance neutrophil transcriptomics[28] and functional studies for the noninvasive monitoring of immunotherapy.

## Methods

**Healthy donor and clinical blood specimens**. All healthy donor and patient blood specimens were acquired and handled according to the protocols approved by the Massachusetts General Hospital (MGH) Institutional Review Board. Whole blood from healthy donors, who were not taking medications including those known to affect platelet functions (e.g., nonsteroidal anti-inflammatory drugs or aspirin) within 48 h prior to phlebotomy, was obtained at MGH or purchased from Research Blood Components (Brighton, MA). Whole blood from a total of 22 cancer patients who were receiving care at the MGH Cancer Center was obtained in one or more occasions. All blood was drawn into ACD-A tubes (BD Vacutainer; 8.5 mL) except for the anticoagulant comparison experiment (Supplementary Fig. 1) in which $K_2$EDTA tubes (BD Vacutainer; 10 mL) were also used. Blood specimens were held at RT (20–25 °C) and were processed as soon as possible. For the majority of the experiments the blood was used within 1–3 h, including transport time from the clinic. Blood specimens that were older than 4 h were discarded. As such, the term "fresh blood" refers to blood used within 4 h. For platelet aggregometry, 19–21 G needles were used exclusively for blood collection, and specimens were held without mixing. Otherwise, all blood specimens were collected using 19–23 G needles and gently mixed (HulaMixer; Life Technologies) prior to experimental treatments and storage.

**Platelet inhibitor treatment and blood storage**. Untreated blood was used without any modifications. For treatment with glycoprotein IIb/IIIa inhibitors, concentrated aqueous stock solutions of tirofiban (0.1 mg mL$^{-1}$; Sigma) and eptifibatide (10 mg mL$^{-1}$; Tocris) were added to fresh blood at final concentrations of 0.5 and 20 or 50 μg mL$^{-1}$, respectively. Blood was then either processed immediately or stored undisturbed in 4 °C or RT (20–25 °C), protected from light, for 24–72 h. After storage, cold-stored blood was allowed to equilibrate to RT before mixing and further processing. No further warming was performed on stored blood specimens and all blood processing took place under RT and standard laboratory

conditions. Where applicable, EDTA (Ambion) was added to blood at final concentrations of 2–5 mM and incubated for 15–75 min prior to experimental assays.

**Blood smears and Wright–Giemsa staining**. Peripheral blood smears were prepared using a wedge technique according to the standard procedures. Briefly, a drop (~ 6 μL) of blood was pipetted onto a glass slide and evenly smeared using a second slide, air dried, fixed with 100% methanol, dipped in Wright–Giemsa stain (Sigma) for 30 s, and rinsed gently with distilled water before air drying. Slides were mounted with Permount (Fisher) before imaging using a Nikon Eclipse 90i microscope with a Nikon DS-Ri1 color camera (12 bit; 1280 × 1024 resolution) and a Nikon 100× Apo VC 100×/1.40 oil objective. About 100 random RBCs were counted per sample and classified as either echinocytes (based on distinct thorny projections) or healthy RBCs.

**Imaging flow cytometry**. Cell viability, leukocyte activation, and platelet–leukocyte adhesion were studied using the ImageStream$^X$ Mark II imaging flow cytometer (Amnis Corporation) equipped with a 40× objective, 6 imaging channels, and 405, 488, and 642 nm lasers. Whole blood was diluted 1:33 in RPMI 1640 medium (supplemented with 10 mM HEPES; Life Technologies) and stained with the following where applicable: calcein blue, AM (10 μM; Life Technologies); Pacific Blue-conjugated CD41 antibody (1:100; clone HIP8; BioLegend, Cat# 303713); CellEvent Caspase-3/7 Green Detection Reagent (5 μM; Life Technologies); Alexa Fluor 488-conjugated CD11b antibody (1:500; clone ICRF44; Stemcell Technologies, Cat# 60040AD); PE-conjugated CD66b antibody (1:150; clone G10F5; Stemcell Technologies, Cat# 60086PE); PE-conjugated EpCAM antibody (1:250; clone VU1D9; Cell Signaling Technology, Cat# 8995s); PE-CF594-conjugated CD45 antibody (1:666; clone HI30; BD Biosciences, Cat# 562279); and DRAQ5 (1 μM; Cell Signaling Technology). Single cells were gated using the nuclear stain DRAQ5. Viable cells were defined as calcein-positive but caspase-3/7-negative.

**Impedence aggregometry**. Platelet aggregation experiments were performed in whole blood using electrical impedance on a two-channel Chrono-Log 700 Series Whole Blood/Optical Lumi-Aggregometer and analyzed using AGGRO/LINK8 software. Whole blood was incubated at 37 °C for 5 min and each sample was run for 10 min after addition of the agonist with a stir bar spinning at 1200 r.p.m. The agonists collagen (Chrono-Log Corporation), ristocetin (Chrono-Log Corporation), and thrombin (Aniara) were used at concentrations of 3 μg mL$^{-1}$, 1.5 mg mL$^{-1}$, and 0.25 U mL$^{-1}$, respectively. Platelet response was quantified by the area under the impedance curve at 6 min after the onset of aggregation (i.e., correcting for the lag time between agonist addition and aggregation).

**Microfluidic isolation of spiked cells and patient CTCs**. LNCaP cells (American Type Culture Collection, Cat# CRL-1740), an androgen-sensitive prostate adenocarcinoma cell line, were authenticated by short tandem repeat profiling and tested negative for mycoplasma contamination. The cells were routinely cultured at 37 °C in 5% $CO_2$ with RPMI 1640 medium supplemented with 10% fetal bovine serum and 1% penicillin–streptocymin (all from Life Technologies). For microfluidic isolation of spiked cells and CTC-iChip enumeration experiments, CellTracker (Life Technologies)-labeled LNCaP cells were trypsinized, resuspended in phosphate-buffered saline (PBS) and passed through 30 μm filters (Miltenyi) to remove aggregates, and spiked into healthy donor blood at 2000 cells per mL of blood. Patient blood specimens were used without spiking. The blood was then divided into appropriate volumes (5–6 mL for spiked cells experiments and 4–8 mL for patient blood), treated with tirofiban where indicated, and immediately stored or processed for microfluidic CTC isolation using the CTC-iChip. As part of the platelet stabilization cocktail treatment (tiro-EDTA), EDTA was added to the blood sample (3 mM final concentration) and incubated for 15 minutes prior to CTC-iChip processing.

CTC isolation was performed as described previously[24]. Briefly, whole blood was incubated with biotinylated antibodies to CD45 (clone HI30; Thermo Fisher Scientific), CD16 (clone 3G8; BD Biosciences), and CD66b (clone 80H3; Novus Biologicals), followed by a second incubation with streptavidin-coupled Dynabeads (Invitrogen), and loaded into a pressurized syringe for processing with the CTC-iChip. The above antibodies were biotinylated by the respective vendors through custom orders. The CTC-iChip first separates nucleated cells from other blood components (i.e., "debulking" by removing plasma, platelets, RBCs, and free beads) by deterministic lateral displacement, before aligning nucleated cells in a single file by inertial focusing and magnetically deflecting bead-labeled leukocytes under continuous flow. Highly enriched CTCs are collected in the product outlet in PBS containing 1% (w/v) Pluronic F-68 (Sigma). Spiked cells were enumerated by counting CellTracker-positive cells in two Nageotte chambers (Hausser Scientific). Leukocytes were enumerated as nucleated cells that were not spiked cells.

**RNA extraction and RNA integrity assessment**. Total RNA was extracted from the CTC-iChip product using standard protocols and the Qiagen RNeasy Plus Micro Kit. Briefly, cells were spun down (300 × g, 5 min) and resuspended in 200 μL of RNAlater (Ambion), flash-frozen in liquid nitrogen, and stored at −80 °C until further processing. RNA concentrations were determined using a NanoDrop

UV-VIS instrument and the ratio of absorbance at 260/280 nm was used as an indicator of RNA purity (A260/280 ratio = ~ 2). RNA quality and RIN values were determined using the Bioanalyzer (Agilent Technologies) per standard protocols.

**Single-cell RT-qPCR**. LNCaPs were cultured in RPMI 1640 medium containing 10% charcoal-stripped FBS (Invitrogen) and treated with R1881 (Perkin-Elmer)[67] before spiking into healthy donor blood. Single LNCaP cells from the CTC-iChip product were micromanipulated using a 20 μm transfer tip (Eppendorf)[13]. Picked cells were placed into individual PCR tubes containing 5 μL RNA lysis buffer and flash-frozen in liquid nitrogen[13]. cDNA synthesis was performed using the Superscript VILO cDNA synthesis kit and the T4 Gene 32 Protein (New England Biolabs, Cat#M0300S). The TaqMan PreAmp Master Mix (Applied Biosystems) was used for pre-amplification (20 cycles) with a stock solution containing all primer sets (100 nM per primer, Supplementary Table 2). Following pre-amplification, samples were exposed to brief digestion with exonuclease I (37 °C for 30 min and 80 °C for 15 min; New England BioLabs) and each sample was diluted fivefold with DNA Suspension buffer (TEKnova). Sample and assay mixtures were loaded onto a 48.48 Dynamic Array integrated fluidic circuit for standard amplification per manufacturer's instructions (Biomark, Fluidigm). The expression level of each gene is presented as fold change relative to the average at 0 h.

**RNA sequencing and bioinformatic analyses**. Using the SMARTer Ultra Low Input RNA Kit for Sequencing—v4 kit (Clontech Laboratories), RNA (~ 1 ng) was converted to cDNA and amplified per the manufacturer's protocol. Briefly, ERCC RNA Spike-In Mix (1 μl of a 1:50,000 dilution; Life Technologies) was added to each sample, and first strand synthesis used a poly-dT primer (3′-SMART CDS primer II A) followed by extension/template switching by reverse transcriptase. After 18 cycles of second strand synthesis and amplification, cDNA was purified with a 1× Agencourt AMPure XP bead cleanup (Beckman Coulter), and the Nextera XT DNA Library Preparation kit (Illumina) was used for sample fragmentation/barcoding. Fragmentation used 1 ng of purified cDNA, followed by 12 cycles of amplification prior to unique dual-index barcoding. After the PCR product was purified with a 1.8× Agencourt AMPure XP bead cleanup, the eluted cDNA did not undergo the bead-based normalization step in the Nextera XT protocol. Validation of the library was performed by qPCR (KAPA SYBR FAST Universal qPCR kit, Kapa Biosystems) before the individual libraries were pooled at equal concentrations. The final pool concentration was determined using the KAPA SYBR FAST Universal qPCR Kit. The pooled libraries were sequenced on the NextSeq 500 platform run at 75 bp paired-end reads (300 cycle kit).

Bioinformatic processing combined the paired-end reads and alignment to the hg38 reference genome using the default setting of the aligner (http://genome.ucsc.edu using the STAR v2.4.0 h aligner). Reads that either mapped to multiple locations or did not map were discarded. The MarkDuplicates tool (picard-tools-1.8.4) was used to mark and remove duplicate reads. Uniquely aligned reads were counted using htseq-count (intersection -strict mode) using a *Homo sapiens* annotation table (GRCh38.79.gtf, http://www.ensembl.org). The raw counts were then imported into R for further analysis. RNA-sequencing expression profiling yielded an average of ~ 12.4 million reads per sample. RPM and log2-transformed values followed standard procedures. The results were plotted as a heatmap using the pheatmap function in R.

**ddPCR detection of AR-V7 mRNA**. For this patient cohort, we randomly selected five patients with unknown AR-V7 status in addition to two patients whose CTCs were known to be AR-V7-positive (from our recent study[13]). RNA from the CTC-iChip product was reverse-transcribed using the First-Strand Synthesis SuperMix protocol (Invitrogen, Cat# 11752). Following reverse transcription, cDNA was combined with Supermix (BioRad) and AR-V7 primers, as per the manufacturer's instructions. ddPCR was performed using an AutoDG automated droplet generator (C1000 Touch Deep-well thermocycler, QX200 plate reader).

**Statistical analyses**. Numerical data are expressed as mean ± s.d. Pair-wise comparisons used two-tailed *t*-test. Comparisons of grouped data used one-way ANOVA followed by Tukey's post test for multiple comparisons. Comparisons of echinocyte formation over storage time and conditions were tested with two-way ANOVA followed by Tukey's post test. Comparisons of single-cell gene expression levels over storage time were tested with two-way ANOVA followed by Dunnett's post test using 0 h as the control. For RNA sequencing, the RPM of fresh vs. stored blood for each cancer-specific target (40 total) was compared using a paired two-tailed *t*-test. A paired test is chosen here because it is more effective in detecting changes as a result of storage when paired data (i.e., 0 h vs. storage) for each patient is available. In our study in which the desired outcome of preservation is to detect no differences, a paired test represents a more stringent test than an unpaired test. All tests were performed with Prism 7 (GraphPad).

**Data availability**. All RNA-sequencing data have been deposited in NCBI GEO under accession number GSE104209. All other data are available within the Article and Supplementary Information, or available from the corresponding authors upon request.

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

## Acknowledgements

We thank Octavio Hurtado, John Walsh, Jon Edd, David Lewis, Stefan Herrara, Kendall Williams, Cindy Angpraseuth, Kate Broderick, Aaron Shank, Coleman Hoff, Gregory Rose, Mary Margaret Crowley, and Caitlin Tompkins for experimental assistance; Charles Vanderburg (Advanced Tissue Resource Core) for helpful discussions and use of the Agilent Bioanalyzer; and Ben Wittner for help in biostatistics and RNA-sequencing analysis. We also thank healthy volunteers and patients who donated blood specimens. This work was supported by funding from the US National Institutes of Health (NIH) P41 BioMEMS Resource Center (EB002503; M.T.), NIH National Institute of Biomedical Imaging and Bioengineering (EB012493; M.T.), NIH National Institute of Neurological Disorders and Stroke (NS045776; S.L.S.), the Howard Hughes Medical Institute (D.A.H.), the Prostate Cancer Foundation (D.T.M.), the US Department of Defense (D.T.M.), the Burroughs Wellcome Trust (D.T.T.), the National Science Foundation (PHY-1549535; D.T.T.), the Lustgarten Foundation (D.T.T.), the Warshaw Institute for Pancreatic Cancer Research (D.T.T.), and the Verville Family Pancreatic Cancer Research Fund (D.T.T.). S.N.T. holds a Postdoctoral Fellowship by the Natural Sciences and Engineering Research Council of Canada (NSERC).

## Author contributions

K.H.K.W. and S.N.T. designed and performed research, analyzed data, and wrote the manuscript. D.T.M., V.T., R.D.S. and D.T.T. assisted with data analysis and manuscript preparation. K.L.M., L.D.B., T.R.C., C.J.S., E.C.T., K.D.V., E.S.E. and H.M.P. designed and performed experiments and analyzed data. L.V.S. assisted in clinical studies. D.A.H., S.M., S.L.S. and M.T. designed and supervised the research and wrote the manuscript.
