## [Peer Review File · Nature Communications]

Reviewers' comments:

Reviewer #1 (Remarks to the Author):

This is an extremely well-designed, meticulously implemented, and important paper that will certainly influence the scientific community that studies CTCs. Multiple important aspects of blood sample preservation were addressed in a detailed, step-by-step manner prior to microfluidic isolation of CTCs. The optimized preservative should enable the performance of relevant clinical trials that would require blood and live CTC preservation during shipping. As such, this manuscript strongly deserves publication.

Several comments/questions:

1. While granulocyte and other cell viability in healthy volunteer blood was tested and shown in Fig 1, comparisons of CTC viability were not shown for the CTCs themselves processed after different storage times. CTC viability equivalency was inferred by the RIN values at 24 (n=4), 48 hrs (n=5), and 72 hrs (n=1) on pooled CTCs from patients. But as future CTC propagation may depend on capturing as high a number as possible of viable CTCs, the percentage of viable CTCs at each time period would be helpful. This could easily be determined using a calcein/ethidium homodimer-1 live/dead cell viability assay (or even using the apoptosis assay shown in Fig 1) with replicate blood samples stored at times 0, 24, 48, and 72 hrs. How this percentage changes over time could impact best storage/transport times for the specific downstream application of CTC culture, mentioned in the discussion, instead of assuming that 0, 24, and 48hrs storage are equivalent. If these new experiments will take too long due to patient sample acquisition (and significantly delay publication of this important paper), I would then suggest specifically mentioning in the discussion that further determination of percentage viable CTCs at different storage times will be optimized in the future for use in CTC propagation experiments.

2. It was not clear to me in Fig 5C what threshold for RPM relative fold change is considered acceptable, given that 0 = no change would be optimal (note that the article referenced in Methods, Ref 57, used log₂ fold change (FC) >1 for showing increased gene expression of single vs. clustered CTCs). In the RPM heatmap in Fig 5B, both AR and KLK2 reads appear to differ at 0 hrs and 48hrs for patient #3.2 (falsely elevated expression?), although a bimodal scale for gene detection/no detection was used to calculate concordance (Supplementary Table S2). Is this a fair comparison measure rather than using relative gene expression? Or was this done because of the many issues involved in analyzing RNASeq data?

See, but don't necessarily cite:

A survey of best practices for RNA-seq data analysis.

Conesa A, Madrigal P, Tarazona S, Gomez-Cabrero D, Cervera A, McPherson A, Szczeńniak MW, Gaffney DJ, Elo LL, Zhang X, Mortazavi A. *Genome Biol.* 2016 Jan 26;17:13. doi: 10.1186/s13059-016-0881-8. PMID: 26813401

Translating RNA sequencing into clinical diagnostics: opportunities and challenges.

Byron SA, Van Keuren-Jensen KR, Engelthaler DM, Carpten JD, Craig DW. *Nat Rev Genet.* 2016 May;17(5):257-71. doi: 10.1038/nrg.2016.10. PMID: 26996076

3. Did any of the four prostate cancer patients have the AR-V7 variant mentioned on p.12, and if so, was there any change in its expression in captured CTCs after different blood sample preservation times?

See (using a different CTC capture method):

CTC-mRNA (AR-V7) Analysis from Blood Samples-Impact of Blood Collection Tube and Storage Time.

Luk AWS, Ma Y, Ding PN, Young FP, Chua W, Balakrishnar B, Dransfield DT, Souza P, Becker TM. *Int J Mol Sci.* 2017 May 12;18(5). pii: E1047. doi: 10.3390/ijms18051047. PMID: 28498319

In summary, this paper is outstanding and should be accepted after answering the minor questions above.

Reviewer #2 (Remarks to the Author):

This study examined the effect of a novel non-fixative, cold-storage processing solution for the treatment of whole blood prior to isolation of circulating tumor cells using a microfluidic device. Addition of platelet activation inhibitors to cold-stored whole blood with the pre-processing addition of a calcium chelators (EDTA) was shown to result in improved storage stability of cells, separation yield of CTCs and stability / quality of cell RNA. Transcriptome analysis of prostate cancer patient samples was used to demonstrate a good concordance in the detection of cancer specific genes following 24-48 h of cold storage.

Major Comments:

1. Bulk RNA analysis of RT stored and/or cold-stored without Tiro/EDTA would provide a more robust evaluation of the beneficial effect of the storage technology on clinical samples. A

comparison only with fixed samples limits the demonstration of practical utility of the Cold w/ Tiro/EDTA treatment. The addition of these two control treatment groups should be added to Figure 4.

2. There are a wide number of chemical agents that have been shown to inhibit platelet activation (anti platelet therapies) including acetylsalicylic acid, dipyridamole, cilostazol, ticlopidin and clopidogrel, abciximab, eptifibatide) besides the tirofiban / eptifibatide used in this study. It is unclear why none of these additional compounds were not evaluated and whether the observed effects would have been similar had any one of the other platelet activation inhibitors had been used. Is physical targeting of the GpIIb/IIIa complex important in maintaining the rheological properties of the platelets so that they are sufficiently separated on the microfluidic device? A further rationale for the selection of the specific platelet activation inhibitor should be provide.

Minor Comments:

1. p. 6, line 3 - It is suggested that ACD-anticoagulated blood was the "optimal method" for maintaining whole blood quality at hypothermic temperatures, but only EDTA was used as a comparator. There are many other whole blood storage solutions available commercially which could have been evaluated. It is inappropriate to state that this is the "optimal method" when no formulation optimization efforts were undertaken. This statement should be revised to reflect that ACD-anticagulated blood simply provided acceptable maintenance of whole blood quality.

2. It is not clear from the data presented whether cold storage of the whole blood alters the rheological properties of the cells sufficiently to reduce their separation on the microfluidic device. It has been reported (Barshtein et al. Transfusion, 2008) that cold treatment can alter the deformability and rheology of red blood cells. Are there changes that need to be made to the microfluidic CTC-iCHIP to accommodate for the effects of cold storage on the CTC separation? Further details on sampling handling (ie. warming) of the cold stored and RT-stored whole blood prior to processing should be included.

3. The impact of temperature on platelet activation and response to agonists is very well described in the literature. The untreated platelet data reported in Figure 2 is expected based on prior published works. The detailed description of the results in the manuscript (p. 6-7) is not required and could be shortened.

4. Efforts should be taken to more clearly define when reference is being made to supplemental figures and not to the manuscript figures. It was unclear when referencing Fig 1C, for example, if the reference was being made to the supplemental material or not. Please correct.

5. The term "fresh blood" is used when referring to control groups. As fresh is an arbitrary term,

it would be more informative if the actual age of the control whole blood at the time of testing is provided.

Responses to Reviewers:

Comments/questions by Reviewer #1:

- 1. While granulocyte and other cell viability in healthy volunteer blood was tested and shown in Fig 1, comparisons of CTC viability were not shown for the CTCs themselves processed after different storage times. CTC viability equivalency was inferred by the RIN values at 24 (n=4), 48 hrs (n=5), and 72 hrs (n=1) on pooled CTCs from patients. But as future CTC propagation may depend on capturing as high a number as possible of viable CTCs, the percentage of viable CTCs at each time period would be helpful. This could easily be determined using a calcein/ethidium homodimer-1 live/dead cell viability assay (or even using the apoptosis assay shown in Fig 1) with replicate blood samples stored at times 0, 24, 48, and 72 hrs. How this percentage changes over time could impact best storage/transport times for the specific downstream application of CTC culture, mentioned in the discussion, instead of assuming that 0, 24, and 48hrs storage are equivalent. If these new experiments will take too long due to patient sample acquisition (and significantly delay publication of this important paper), I would then suggest specifically mentioning in the discussion that further determination of percentage viable CTCs at different storage times will be optimized in the future for use in CTC propagation experiments.*

We greatly appreciate the reviewer's questions regarding the impact of storage on CTC viability. Indeed our achievement in whole blood stabilization opens up opportunities in *ex vivo* CTC propagation post-storage. As the reviewer has pointed out, collecting such data from patient samples will require significant time investment, especially considering the heterogeneity and variability in the number,

biology, and complexity of culture for these rare cells. To address the reviewer's comments, we have performed experiments using spiked LNCaP cells in both healthy donor and patient blood and used our calcein/caspase assay to quantify spiked cell viability. We found that cold storage for up to 72 hours did not negatively affect tumor cell viability, consistent with our results on RNA quality and sequencing. We have added these results in the revised manuscript (Supplementary Fig. 6; page 17, line 367-375).

2. *It was not clear to me in Fig 5C what threshold for RPM relative fold change is considered acceptable, given that 0 = no change would be optimal (note that the article referenced in Methods, Ref 57, used log₂ fold change (FC) >1 for showing increased gene expression of single vs. clustered CTCs). In the RPM heatmap in Fig 5B, both AR and KLK2 reads appear to differ at 0 hrs and 48hrs for patient #3.2 (falsely elevated expression?), although a bimodal scale for gene detection/no detection was used to calculate concordance (Supplementary Table S2). Is this a fair comparison measure rather than using relative gene expression? Or was this done because of the many issues involved in analyzing RNASeq data?*

We thank the reviewer for bringing up these insightful questions of gene expression and analysis methods. We address this question in three parts as follows:

(2.1) It was not clear to me in Fig 5C what threshold for RPM relative fold change is considered acceptable, given that 0 = no change would be optimal (note that the article referenced in Methods, Ref 57, used log₂ fold change (FC) >1 for showing increased gene expression of single vs. clustered CTCs).

We thank the reviewer for addressing the concept of threshold for RPM relative fold change; it is an important and critical consideration. As a result of this astute comment from the reviewer, we have performed further review of our methods and improved our approach to handle our unconventional statistical situation. Dr. Ben Wittner, the bioinformatics specialist who lead the statistical analysis efforts in Aceto *et al.* (ref. 57), assisted with our revised analysis. Below, we describe in detail the rationale for our original and updated approaches.

In conventional hypothesis testing, it is considered worse to make a type I error in comparison to a type II error. As a result, a "more stringent", or conservative, statistical analysis would make it more difficult to detect real differences and thus would lower the chances of making a type I error. In other words, *a stringent statistical test for detecting changes favors the null hypothesis*. The practice of higher stringency is well-established and preferred in the biomedical sciences. For instance, in Aceto *et al.* (ref. 57) referenced by the reviewer, the goal was to identify differences across the whole transcriptome which requires correction of multiple comparisons. When working with such large data sets, a proportion of gene expression changes are likely false positives; however, overly stringent statistical methods commonly used in relatively small datasets (e.g. Bonferroni) can dramatically increase type II errors. As a result, setting a threshold based on the dataset is a reasonable method to control for multiple comparisons (and type I errors) while maintaining high statistical power.

In our unconventional situation however, in which the desired outcome is to observe *no difference* in gene expression as a result of storage (i.e., to accept the null hypothesis), a more stringent test in the traditional sense in fact works in our favor. It was because of this reason that our original approach was to use a simple *t* test, which works *against* our desired outcome by being easier in detecting differences—and thus could be considered more stringent in our special case. With this method, we only found significant difference in one gene when comparing preserved against fresh blood.

However, the issues regarding the use of conventional hypothesis testing to analyze "no difference" has been acknowledged by biostatisticians. Often, it is of interest to ascertain whether new treatments or medications are at least as effective as existing gold standards. In these cases, a different statistical approach has been developed which are termed "equivalence" or "non-inferiority" trials. To perform an

equivalence analysis with our results, one would require a clinically justified margin of acceptable difference; then, if the 95% confidence interval of the mean in the treatment group (i.e., post-storage) falls within this equivalence margin, then one can conclude that the treatment is equivalent. Unfortunately, these are only suited for large-scale clinical trials which are beyond the scope of the present study.

In addition, another limitation to using this approach is that the equivalence margin is determined by biologically and clinically meaningful outcomes. In the burgeoning field of precision medicine, such quantitative information remains to be determined and validated. To this end, future studies that take advantage of next-generation sequencing technologies will be critical to establish the criteria for biomarker preservation.

Acknowledging the above complexities, we implement the following to improve our statistical analysis as well as data presentation and discussion.

1. We have changed the statistical analysis from a *t* test to a paired *t* test. Because of considerable inter-patient variability, a paired test is more appropriate to detect changes specific to storage and thus more stringent to our application. With this method and with updated RNA-Seq data (see below), we detect significant difference in 1 gene out of 40. We have edited the text to explain this choice and updated our results (page 11, line 234-236; page 26, line 570-575). We have also added the *p* values to Figure 5 for clarity.
2. We add a discussion point which acknowledges that preservation of gene expression will have to be validated with a larger cohort of patients. We also note that an equivalence trial is the most appropriate and look forward to such a study (page 16-17, line 355-358).

(2.2) In the RPM heatmap in Fig 5B, both AR and KLK2 reads appear to differ at 0 hrs and 48 hrs for patient #3.2 (falsely elevated expression?)

The reviewer has made another excellent point. Indeed, we sometimes observe an increase in gene expression after storage. Instead of falsely elevated expression, we believe that CTCs may be selectively upregulating certain transcripts. For example, in our data from single LNCaP cells presented in Figure 4, we found that fibronectin was upregulated (it was the only target which showed a significant increase in gene expression). We reasoned that since LNCaP cells are adapted to adherent culture conditions prior to being transferred to a non-adherent suspension for several days in blood, the increase in fibronectin expression may reflect a compensatory response as an attempt to assemble an extracellular matrix. Together with results showing that tumor cells remain viable during whole blood storage (Supplementary Figure 6), we speculate that these changes in gene expression are real and that CTCs are responsive to the environment. We have edited the text to discuss these points (page 11, line 236-238).

(2.3) a bimodal scale for gene detection/no detection was used to calculate concordance (Supplementary Table S2). Is this a fair comparison measure rather than using relative gene expression? Or was this done because of the many issues involved in analyzing RNASeq data?

We thank the reviewer for the question on concordance (now Supplementary Table 1 in the revised manuscript). The data used to generate this binary concordance table is based on the same RNA sequencing data set which is also presented in Figure 5B as reads per million (RPMs). Our intention was to provide two different ways to interpret the same data, because we think both expression levels and concordance are useful information. Our rationale for presenting a binary concordance table is because certain diagnostic markers are clinically relevant simply based on its presence or absence regardless of their absolute quantities.

An important example relevant to the present study is AR-V7, which has been shown to identify metastatic castration-resistant prostate cancer patients who are resistant to AR signaling inhibitors

enzalutamide and abiraterone (Antonarakis et al., *NEJM*, 2014) and these patients may respond better to taxane therapy instead (Antonarakis et al., *JAMA Oncol.*, 2015). Importantly, the presence of AR-V7 is predictive regardless of its quantity, and a binary “presence or absence” may be sufficient for clinical utility. Also, studies that investigate the clinical relevance with regard to transcript *copy numbers* are limited. One study demonstrated that KLK2 and KLK3 (PSA) mRNAs were strongly correlated with CTC count for patients with high CTC burden (>15 CTCs/7.5 mL blood using CellSearch; Helo et al., *Clin. Chem.*, 2009). Yet, association with survival was again based on a binary scale of mRNA positivity.

In considering the reviewer’s point on this matter, we have edited the manuscript to elaborate on these details (page 17, line 359-366). Further, we explicitly state in the revised manuscript that our concordance metric is based on a binary scale (Supplementary Table 1; page 2, line 45; page 11, line 240; page 12, line 258; page 17, line 362).

3. *Did any of the four prostate cancer patients have the AR-V7 variant mentioned on p.12, and if so, was there any change in its expression in captured CTCs after different blood sample preservation times?*

We very much appreciate this comment, and we completely agree that this data will significantly strengthen the present study. The main challenge with detecting AR-V7 is that its expression level is much lower relative to the other cancer-specific genes (Antonarakis et al., *NEJM*, 2014 & Miyamoto *et al.*, *Science*, 2015). Therefore we used a two-pronged approach as follows: 1) we re-sequenced our current patient cohort (and added one more patient sample which was preserved for 72 hours) with ~12-fold more depth; 2) we quantified AR-V7 mRNA copy numbers using droplet digital PCR (ddPCR) which has a higher detection sensitivity. Our results indicate that indeed sequencing with ~12-fold more depth was insufficient to detect AR-V7; this result is consistent with our estimate using data from Miyamoto *et al.*, (*Science* 2015) which suggests that 10~100 fold more depth may be required. However, ddPCR was able to reliably detect AR-V7 from our patient cohort with 100% binary concordance. We have updated these results in the revised manuscript (Figure 5C; page 2, line 44; page 12, line 247-258).

Comments by Reviewer #2

Major comments:

1. *Bulk RNA analysis of RT stored and/or cold-stored without Tiro/EDTA would provide a more robust evaluation of the beneficial effect of the storage technology on clinical samples. A comparison only with fixed samples limits the demonstration of practical utility of the Cold w/Tiro/EDTA treatment. The addition of these two control treatment groups should be added to Figure 4.*

We very much appreciate this comment and agree that adding these control experiments with patient blood will highlight the utility of our preservation method. We performed additional experiments using RT- and cold-stored patient blood without tiro-EDTA, and found that the RNA quality of the CTC product is extremely poor. These results are added to the revised manuscript (Supplementary Figure 4; page 10-11, line 223-227) and included below. In instances where the RIN values could not be computed because of poor RNA quality, a RIN value of “1” (the lowest possible RIN) was assigned for easy visualization.

2. *There are a wide number of chemical agents that have been shown to inhibit platelet activation (anti platelet therapies) including acetylsalicylic acid, dipyridamole, cilostazol, ticlopidin and clopidogrel, abciximab, eptifibatide) besides the tirofiban / eptifibatide used in this study. It is unclear why none of these additional compounds were not evaluated and whether the observed effects would have been similar had any one of the other platelet activation inhibitors had been used. Is physical targeting of the GPIIb/IIIa complex important in maintaining the rheological properties of the platelets so that they are sufficiently separated on the microfluidic device? A further rationale for the selection of the specific platelet activation inhibitor should be provide.*

We thank the reviewer for this insightful question about other anti-platelet agents. Indeed, we did perform initial experiments using a variety of platelet inhibitors. We ultimately finalized on tirofiban and eptifibatide because they were extremely effective in platelet inhibition in all our assays (platelet count, aggregometry, microfluidic processing, and CTC sorting), and thus were deemed sufficient for this work. Other anti-platelet agents tested include acetylsalicylic acid (0.5-2 mM) and clopidogrel (10-100 µg/mL) as the reviewer suggested, as well as a p-selectin inhibitor KF 38789 (5-40 µg/mL). We used our microfluidic filter device (Supplementary Figure 2) but found no functional benefits on processing stored blood. We did not test dipyridamole and cilostazol due to potential undesired biochemical effects resulting from phosphodiesterase inhibition in other cell types. We also did not test ticlopidin and abciximab, although we expect the latter would also be effective because it is also a GPIIb/IIIa inhibitor similar to tirofiban and eptifibatide. We attribute their effectiveness to the targeted inhibition of GPIIb/IIIa, which is the most abundant receptor on platelets and is required for fibrinogen binding, platelet aggregation, and thrombus growth. As the reviewer has pointed out, GPIIb/IIIa inhibitors allow platelets to remain separated and thereby enable clog-free microfluidic processing. We have edited the discussion to include these details (page 15, line 321-328).

Minor comments:

1. *p. 6, line 3 - It is suggested that ACD-anticoagulated blood was the "optimal method" for maintaining whole blood quality at hypothermic temperatures, but only EDTA was used as a comparator. There are many other whole blood storage solutions available commercially which could have been evaluated. It is inappropriate to state that this is the "optimal method" when no formulation optimization efforts were undertaken. This statement should be revised to reflect that ACD-anticoagulated blood simply provided acceptable maintenance of whole blood quality.*

We agree with the reviewer and have edited the text accordingly (page 5, line 103; page 6, line 119).

2. *It is not clear from the data presented whether cold storage of the whole blood alters the rheological properties of the cells sufficiently to reduce their separation on the microfluidic device. It has been reported (Barshtein et al. Transfusion, 2008) that cold treatment can alter the deformability and rheology of red blood cells. Are there changes that need to be made to the microfluidic CTC-iCHIP to accommodate for the effects of cold storage on the CTC separation? Further details on sampling handling (ie. warming) of the cold stored and RT-stored whole blood prior to processing should be included.*

We thank the reviewer for this question. We are aware of storage-induced changes in RBC rheology. According to the reference provided by the reviewer and an earlier study (Berezina et al. *J. Surg. Res.*, 2002), changes in RBC shape and deformability in cold storage were insignificant until after two weeks. These results are consistent with our CTC-iChip data regarding RBC contamination in the product (Figure 3B), indicating that cold storage for up to 3 days (with tiro-EDTA) does not result in compromised microfluidic RBC removal. However, the presence of blood clots in the CTC-iChip (i.e., cold without tiro-EDTA) does result in significantly more RBC contamination by interfering with microfluidic sorting (Figure 3A). These data further demonstrate the combined benefits of hypothermic preservation and

platelet stabilization—that no changes are necessary in terms of microfluidic operation and blood handling. Stored blood specimens are simply equilibrated to room temperature before standard CTC-iChip processing procedures. We have edited the manuscript to discuss these details (page 14, line 305-307; page 15, line 333-334; page 20, line 439-441).

3. *The impact of temperature on platelet activation and response to agonists is very well described in the literature. The untreated platelet data reported in Figure 2 is expected based on prior published works. The detailed description of the results in the manuscript (p. 6-7) is not required and could be shortened.*

We thank the reviewer for the suggestions. We have shortened this description and cited the relevant references (page 6, line 134-136).

4. *Efforts should be taken to more clearly define when reference is being made to supplemental figures and not to the manuscript figures. It was unclear when referencing Fig 1C, for example, if the reference was being made to the supplemental material or not. Please correct.*

We thank the reviewer for the suggestions. We have edited all references to supplementary figures for clarity.

5. *The term "fresh blood" is used when referring to control groups. As fresh is an arbitrary term, it would be more informative if the actual age of the control whole blood at the time of testing is provided.*

We thank the reviewer for the suggestion. We make every effort to use the blood as soon as possible, and specimens that were older than 4 hours were discarded. Practically speaking, when including transport time from the clinic, we were able to use the blood within 1-3 hours of the initial draw for the majority of the experiments. To include information in the text for the actual age (e.g., accuracy to minutes) would be challenging due to the multiple treatments and assays and the sheer number of samples used for the study. We have included these additional details for the age of our 'fresh' samples in the revised manuscript (page 20, lines 425-428).

We thank the reviewers for their time and insightful comments which have significantly strengthened this work. We hope the revised manuscript is now suitable for publication in *Nature Communications* and we look forward to your decision.

Sincerely,

Mehmet Toner, Ph.D.
Helen Andrus Benedict Professor of Bioengineering
Harvard Medical School and MGH
Director, MGH BioMEMS Resource Center

Shannon Stott, Ph.D.
Assistant Professor, Department of Medicine
Harvard Medical School and MGH
MGH Cancer Center / BioMEMS Resource Center

Reviewers' Comments:

Reviewer #1 (Remarks to the Author):

All questions addressed satisfactorily. Excellent contribution to the field! Recommend acceptance.

Reviewer #2 (Remarks to the Author):

The authors have adequately addressed my comments. I have no further concerns with this manuscript.